# *MGMT* Methylation Is Associated with Human Papillomavirus Infection in Cervical Dysplasia: A Longitudinal Study

**DOI:** 10.3390/jcm12196188

**Published:** 2023-09-25

**Authors:** Boram Choi, Yoojin Na, Min Yeop Whang, Jung Yoon Ho, Mi-Ryung Han, Seong-Woo Park, Heekyoung Song, Soo Young Hur, Youn Jin Choi

**Affiliations:** 1Department of Obstetrics and Gynecology, Seoul St. Mary’s Hospital, College of Medicine, The Catholic University of Korea, 222, Banpo-daero, Seocho-gu, Seoul 06591, Republic of Korea; choiboram0314@gmail.com (B.C.); yoojinna@naver.com (Y.N.); mywhang95@naver.com (M.Y.W.); hojy2000@catholic.ac.kr (J.Y.H.); 2Cancer Research Institute, College of Medicine, The Catholic University of Korea, 222, Banpo-daero, Seocho-gu, Seoul 06591, Republic of Korea; 3Division of Life Sciences, College of Life Sciences and Bioengineering, Incheon National University, Incheon 22012, Republic of Korea; genetic0309@gmail.com (M.-R.H.); swoo1560@naver.com (S.-W.P.); 4Department of Obstetrics and Gynecology, Incheon St. Mary’s Hospital, College of Medicine, The Catholic University of Korea, Incheon 21431, Republic of Korea; songdeng77@naver.com

**Keywords:** cervical dysplasia, human papillomavirus, *MGMT*, methylation

## Abstract

Cervical premalignancy/malignancy, as detected by cervical cytology or biopsy, can develop as a result of human papillomavirus (HPV) infection. Meanwhile, DNA methylation is known to be associated with carcinogenesis. In this study, we thus attempted to identify the association between *MGMT* methylation and persistent HPV infection using an Epi-TOP MPP assay. Integrative analysis of DNA methylation was carried out here using longitudinal cervical cytology samples of seven patients with atypical squamous cells of undetermined significance/low-grade squamous intraepithelial lesion (ASC-US/LSIL). Then, a gene expression analysis using the longitudinal cervical cytology samples and a public database (The Cancer Genome Atlas (TCGA)) was performed. Upon comparing the ASC-US or LSIL samples at the 1st collection and the paired samples at the 2nd collection more than 6 months later, we found that they became hypermethylated over time. Then, using the longitudinal data, we found that the *MGMT* methylation was associated with HPV infection. Moreover, TCGA dataset revealed an association between downregulated *MGMT* mRNA expression and poor overall survival. This decreased *MGMT* mRNA expression was observed to have an inverse relationship with *MGMT* methylation levels. In this study, we found that the *MGMT* methylation level could potentially serve as a valuable prognostic indicator for the transition from ASC-US/LSIL to cervical cancer.

## 1. Introduction

Cervical cancer is a prevalent form of cancer, ranking as the fourth most common type in females and the second leading cause of cancer-related death among women [1,2]. The disease is characterized by the transformation of premalignant to malignant lesions, wherein persistent infection with human papillomavirus (HPV) is the most significant etiological factor [3,4]. Cervical premalignancies/malignancies can be detected through cervical biopsy or cytology. Cervical cytology results are classified based on the PAP class system (Bethesda system). This system classifies squamous cell abnormalities as (1) atypical squamous cells of undetermined significance (ASC-US), (2) atypical squamous cells that cannot exclude high-grade squamous intraepithelial lesion (HSIL) (ASC-H), (3) low-grade squamous intraepithelial lesion (LSIL), (4) HSIL, and (5) squamous cell carcinoma (SCC) [5]. The histological results of the cervical biopsy are reported using either the classical cervical intraepithelial neoplasia (CIN) system or the lower anogenital squamous terminology (LAST) system [6]. It is believed that HSIL cytology is associated with a high risk of ≥CIN 2 (using the CIN system) or HSIL (using the LAST system); therefore, immediate colposcopic examination or tissue biopsy is recommended in such cases [7]. However, the interpretation of LSIL/ASC-US remains controversial because these cases exhibit a wide range of histological results. Although most cases of CIN 2 present spontaneous regression, some are associated with invasive carcinoma. Furthermore, some cases of LSIL/ASC-US progress to HSIL [8]. Given this context, there is a need to identify a biomarker that can predict whether LSIL/ASC-US will progress to HSIL or regress.

More than 200 types of human papillomavirus (HPV) have been identified, of which 40 types are known to infect the genital mucosa, including the cervix [9,10]. Of those, the high-risk HPV types are known to cause cervical precancerous lesions and cancer. These types are present in 99.7% of cervical cancer specimens [11]. High-risk HPV types include 16, 18, 31, 33, 34, 35, 39, 45, 51, 52, 56, 58, 66, 68, and 70 [12]. In an international meta-analysis, it was found that 70% of patients with invasive cervical cancer were infected with HPV 16 (55%) or HPV 18 (15%) [13]. Furthermore, HPV types 58, 52, 45, 33, and 31 in combination accounted for 18% of cases [14]. It was also revealed that the likelihood of developing a cervical malignancy was significantly higher in women with persistent high-risk HPV infection than in those with transient high-risk HPV infection. Another study discovered that the diagnosis of ≥CIN 2 was significantly connected to persistent HPV infection at two time points (during enrolment and at follow-up an average of 5.5 ± 2.5 months later); relative risk, 5.7; 95% CI 2.9–11.3; *p* = 0.001). The study demonstrated that HPV persistence carried a 5.7-fold risk of developing >CIN 2 [15]. In addition, studies have supported the assertion that persistent high-risk HPV infection is a significant factor in the development and maintenance of CIN 3 [16]. While it has been established that persistent infection is the primary cause of cervical cancer, the causes of this persistence and the existence of biomarkers to predict it remain unknown. Previous studies attempted to elucidate the genetic factors that cause the progression of cervical premalignancy to malignancy [9,10,11,12]. For example, Jung et al. described that some genes with somatic mutations (e.g., *PIK3CA, TP53, STK11*, and *MAPK1*) and copy number variations (CNVs) may be associated with cervical cancer progression [17]. Bodelon et al. analyzed CNVs and HPV DNA integration in patients diagnosed with CIN 3 and cervical cancer. They found that the genomic instability index, defined as the proportion of the genome with CNVs, increased with HPV integration and at the transition from CIN 3 to cancer. In addition, they suggested that chromosomal instability may facilitate integration [18]. Jiang et al. also showed that *CCND2* overexpression may be a marker of cervical cancer progression [19].

Epidemiological and molecular studies have supported the assertion that persistent infection with high-risk HPV is required for the development of cervical cancer, in addition to other prerequisite factors [20]. Gene methylation has been implicated in the development of cervical cancer, particularly in the progression of the disease from a premalignant state to a malignant one [21,22]. On the basis of accumulating evidence, it has been suggested that epigenetic silencing of tumor suppressor genes is required for both carcinogenesis and metastasis. During the process of carcinogenesis, which is a multistep process, DNA methylation is one of the earliest and most frequently observed molecular changes [3]. Studies have shown a tendency for the accumulation of DNA methylation, an important epigenetic mechanism for gene silencing, to be associated with disease severity, and this has been supported by findings in cervical premalignancy and malignancy [20,23]. Research has also revealed that a key factor in DNA repair is O6-methylguanine-DNA methyltransferase (*MGMT*), the function of which involves removing mutagenic and cytotoxic adducts from O6-guanine in DNA [24,25]. Specifically, alkyl adducts are removed from the O6 position of guanine in DNA by *MGMT*, leading to the development of resistance against alkylating agents. It has also been shown that *MGMT* is active in the development, progression, and diagnosis of cancer [26]. Indeed, it was shown that approximately 17% (5–92%) of cervical cancers feature loss of expression of *MGMT*. The results also showed an association between cervical premalignant/malignant lesions and hypermethylation of the *MGMT* promoter [25]. Moreover, Kim et al. found that there was an increase in the frequency of the hypermethylation of the *MGMT* promoter in increasingly severe cervical lesions (normal: 2.4%; LSIL: 3.1%; HSIL: 11.9%; SCC: 26.1%; *p* < 0.001). Taking these results together, it seems that the association between *MGMT* methylation and persistent HPV infection could be clarified by analyzing markers of DNA methylation [27].

In the present study, the authors aimed to identify the potential correlation between *MGMT* methylation and persistent HPV infection by analyzing serial cervical cytology samples. The authors utilized The Cancer Genome Atlas (TCGA) database to validate the findings and establish any clinical implications of *MGMT* methylation in the pathogenesis of cervical cancer.

## 2. Materials and Methods

### 2.1. Specimens

A total of 14 samples from 7 patients were used for methylation analysis in this study. We obtained pairs of cervical samples from patients diagnosed with ASC-US/LSIL collected ≥6 months apart. Thus, two DNA samples from each of the seven patients were analyzed (cervical samples at the 1st collection and the ones at the 2nd collection). All of the samples were from a Korean human papillomavirus (HPV) cohort study. Detailed information on that study, including the inclusion/exclusion criteria, is presented elsewhere [28]. All specimens from the seven patients in this study were obtained with appropriate consent. The study also received institutional review board (IRB) approval from Seoul St. Mary’s Hospital College of Medicine, the Catholic University of Korea (KC16TISI0367). All procedures conducted in this study were performed in line with the institution’s ethical standards and with the 1964 Helsinki Declaration and its later amendments.

### 2.2. DNA Extraction and Methylation Analysis

In this study, DNA was extracted from cervicovaginal swabs using a DNeasy Blood and Tissue kit (Qiagen, Valencia, CA, USA) in accordance with the manufacturer’s instructions. DNA samples were stored at −20 °C until analysis.

The methylation status of *MGMT* was evaluated using the Epi-TOP MPP assay (Seasun Biomaterials, Daejeon, Republic of Korea), following the manufacturer’s instructions. The assay used a methylation-specific peptide nucleic acid (PNA) probe that can bind specifically to methylated cytosine and determines the relative percentages of methylation of the targets compared with the amplification rate of an internal control included in the test. In the assay, 5 ng of template DNA was used per reaction. This is a considerably smaller amount of DNA than required in methylation-specific PCR or conventional pyrosequencing, which are followed by bisulfite conversion of nonmethylated cytosine residues. To quantify the degree of methylation in the target area, we used the methylation percent ratio (MPR), which was calculated as follows:MPR = 100/(1 + 2ΔCt target − ΔCt control)

The methylation percent ratio (MPR) (or percent methylation ratio (PMR)) is a standard term routinely used in scientific papers related to methylation analysis. It represents the relative methylation percentage of a target region compared with that of a reference gene. The manufacturer used normal uterine tissues to define *MGMT*’s unmethylated state; MPR ≤ 0.02 was considered to represent an unmethylated state.

For the statistical analysis, the Wilcoxon’s test was used to analyze the association of the pairs of cervical cytology samples obtained >6 months apart from the same individual.

### 2.3. The Cancer Genome Atlas (TCGA) Data Analysis

Correlations between the methylation β-values and mRNA expression levels were analyzed, and 306 TCGA Cervical Squamous Cell Carcinoma and Endocervical Adenocarcinoma (CESC) samples, which matched in terms of the mRNA and methylation data, were included in the analysis. Pearson’s correlation tests were used to examine correlations between methylation and gene expression. Student’s t-tests were performed to test for significance in correlation coefficients. Statistical analyses were conducted in R-4.0.3 and visualized using ggplot2 (https://ggplot2.tidyverse.org/, (accessed on 17 November 2022)) and ggpubr R packages (https://rpkgs.datanovia.com/ggpubr/) (accessed on 17 November 2022). A *p*-value < 0.05 indicated statistical significance.

We used public cervical cancer data to validate our Epi-TOP MPP assay results (methylation assay). TCGA-CESC and GTEx-Cervix Uteri RNA-seq expression data in the transcripts per million (TPM) format and Methylation 450 k Array β-value CESC data were obtained from the UCSC Xena database (http://xena.ucsc.edu/) (accessed on 17 November 2022) [29]. 

## 3. Results

### 3.1. MGMT Methylation and HPV Infection Status in ASC-US/LSIL Patients

This study included seven ASC-US/LSIL patients who were infected with HPV (Table 1). The patients were aged 29–49. All of the patients underwent cervical cytology and HPV tests at the time of the initial sample collection (the 1st collection) and follow-up collection >6 months later (the 2nd collection). In all of the 14 samples at the 1st and 2nd collections, HPV infection was identified. All of the samples at the 1st collection and 2nd collection were persistently infected with the same high-risk HPV types. Among them, the cervical cytology of three patients (Pt1, Pt3, and Pt7) changed from ASC-US/LSIL at the 1st collection to negative for intraepithelial lesions or malignancy (NIL) at the 2nd collection.

The methylation of *MGMT* was analyzed using the Epi-TOP MPP assay, the results of which are shown in Figure 1 and Table 2. This assay selectively amplified methylated copies of *MGMT*, and we determined the methylation status of the target region by calculating the differences in the Ct values (ΔCt) between *MGMT* and the internal control (ACTB). Since the kit specifically amplifies the methylated copies of the target, an unmethylated sample presents a higher ΔCt, while a methylated sample displays smaller Ct value differences. The MPR was calculated from the assay results, and the manufacturer defined the *MGMT* MPR of the unmethylated values as *MGMT* MPR ≤ 0.02. We compared the MPR of the 1st collected sample to that of the 2nd collected sample. A higher MPR indicates that the sample is more methylated. The data demonstrate that all samples with persistent HPV infection were hypermethylated at both the 1st and 2nd collections, regardless of the cervical cytology outcomes at the time of the 2nd collection (defined as unmethylated *MGMT* MPR ≤ 0.02) (see Figure 2).

We investigated whether persistent HPV infection is associated with the *MGMT* methylation status and compared the MPR of the samples collected during the first and second cervical smears. To normalize the MPR level, we subtracted the MPR level of the normal samples (*MGMT* MPRnorm = MPR (cervical samples at the 1st or 2nd collection) − MPR (0.02)). We then compared the *MGMT* MPRnorm of the samples from the 1st collection with that for the samples from the 2nd collection. Our data show a correlation between the *MGMT* methylation status at the beginning of the study (1st collection) and ≥6 months later (2nd collection) (*p* = 0.043) (see Figure 3).

### 3.2. MGMT Methylation and mRNA Expression

We examined the correlations between *MGMT* methylation and mRNA using data from TCGA database. The methylation status of the *MGMT* CpG probe (cg028202904) utilized in the Epi-TOP MPP assay for public data analysis was also assessed. Our findings indicate that the *MGMT* mRNA expression levels were significantly and negatively correlated with methylation at cg028202904 (see Figure 4).

### 3.3. MGMT as a Prognostic Biomarker

Next, to determine whether *MGMT* methylation has any clinical significance, we analyzed the survival of cervical cancer patients using the online tool Gene Expression Profiling Interactive Analysis (GEPIA; http://gepia.cancer-pku.cn) (accessed on 17 November 2022). As seen in Figure 4, we observed a negative correlation between *MGMT* methylation and *MGMT* mRNA expression levels. To further evaluate this clinically, we utilized *MGMT* expression and cervical cancer survival data from TCGA. According to the analysis conducted by GEPIA, a significant association was found between downregulated *MGMT* mRNA expression at the transcriptional level and poor overall survival (*p* = 0.029, log-rank test) in human cervical cancer. However, the association between *MGMT* mRNA expression and disease-free survival (DFS) did not show statistical significance (*p* = 0.33) (refer to Figure 5).

## 4. Discussion

In this study, we examined the potential association between *MGMT* methylation status and HPV infection in ASC-US/LSIL. Our initial findings indicate that the ASC-US and LSIL samples exhibited hypermethylation. Additionally, our longitudinal analysis revealed that *MGMT* methylation was significantly associated with persistent HPV infections in the samples collected more than 6 months apart. Further analysis of TCGA data demonstrated an inverse correlation between *MGMT* methylation and mRNA expression in cervical cancer.

Our data confirm that *MGMT* methylation occurs in precancerous lesions. Several studies have shown an association between the methylation of gene promoters and the development of cervical cancer. Narayan et al. proposed that epigenetic changes in *HIC1*, *RARB*, *DAPK*, and *CDH1* genes may play roles in the development of cervical cancer and act as prognostic indicators [30]. In addition, the silencing of P16 has been demonstrated to induce apoptosis, while P16 methylation has been identified as an early event in cervical carcinogenesis [31]. Moreover, *MGMT*, located on chromosome 10q26.3, encodes an enzyme that plays a role in DNA “suicide” repair [32,33]. This enzyme facilitates the repair of damaged guanine nucleotides via the transfer of the methyl group at the O6 site of guanine to its cysteine residue, thus restoring normal base pairing. Silencing of the *MGMT* gene has been linked to an increased risk of carcinogenesis and susceptibility to therapeutic methylating agents [34]. Loss of *MGMT* expression is primarily regulated by the methylation of *MGMT* promoter CpG islands [35,36]. In this study, we investigated the *MGMT* methylation status of cervical cytology samples and observed hypermethylation in ASC-HS/LSIL samples. Changes in *MGMT* levels were detected in early-stage cervical dysplasia (ASC-HS/LSIL). This indicates the potential significance of *MGMT* methylation as a biomarker for the early detection of cervical cancer. Previous studies have investigated this variable in multiple cancer types including colon cancer [34,35,36,37,38,39,40], neck squamous cell carcinoma [41,42], ovarian cancer [43], and breast cancer [44]. Certain studies have found that the abnormal methylation of *MGMT* was strongly associated with the risk and histological type of cervical cancer [24,45].

The current study is limited by the small sample size. However, in the longitudinal analysis, significant results were obtained. We observed a change in the level of MGMT methylation in seven samples that were followed longitudinally, based on pairs of samples that were obtained more than 6 months apart. Nevertheless, further research is needed to investigate the relationship between the presence and extent of MGMT methylation and the prognosis of cervical pre-cancer and cervical cancer. Although our data could not establish a conclusive association between MGMT methylation and cervical cancer progression because of the small sample size, we attempted to identify any possible correlation using publicly available data from TCGA. TCGA data revealed a significant negative correlation between MGMT methylation levels and mRNA expression in cervical cancer. In addition, our results suggest that this correlation may also have an impact on survival rates. Our data may suggest that MGMT methylation is associated with cervical cancer progression, as a previous longitudinal study showed that *MGMT* promoter methylation is particularly pronounced during tumor progression in oligodendroglioma [46].

We suggest that *MGMT* methylation analysis may be clinically useful along with HPV test. First, *MGMT* methylation may reflect the HPV-infected state. Observational studies performed on a large scale have recently demonstrated that the distribution of HPV genotypes is heterogeneous among women from different ethnic groups [47,48]. A previous study showed that HPV status (HPV types and transient/persistent HPV infection) may change regardless of the PAP cytology results [49], and our data are consistent with this. Furthermore, addressing the specificity of HPV testing, recent studies have demonstrated that HPV-based screening offers results that are at least as reliable, if not superior, when it comes to predicting CIN 2+, CIN 3+, and invasive cervical carcinomas compared with traditional cytology-based methods [50,51]. It is recommended that healthcare providers conduct close observation of patients who test negative on a PAP smear but with HPV infection, regardless of the HPV type, because a PAP smear has a high false-negative rate (27.1%) [52]. Moreover, there is high variability in the reported rates of false negative results (15–65%) [53,54,55,56]. As mentioned above, HPV status may change over time and a PAP smear may produce false-negative results, so in such cases *MGMT* methylation analysis may help identify the patients’ cervical disease status (NIL, precancer or cancer). Second, MGMT methylation might serve as a biomarker for predicting persistent HPV infection. HPV DNA testing and PAP cytology provide information on disease status at the time of analysis but do not offer predictive insight into which patients with high-risk HPV infections will remain persistently infected or progress to cervical cancer. Currently, there are no markers available for predicting HPV infection persistence or the development of cervical cancer. Our study shows an association between MGMT methylation and persistent HPV infection, suggesting that MGMT methylation may have a role as a biomarker for predicting persistence in such cases.

## 5. Conclusions

In this study, it was noted that, over time, ASC-US or LSIL samples with persistent high-risk HPV infections displayed a gradual increase in *MGMT* methylation levels. Additionally, we verified the significance of *MGMT* methylation in the progression of cervical cancer by analyzing public data from TCGA. Our findings suggest that the *MGMT* methylation level could potentially serve as a valuable prognostic indicator for the transition from ASC-US/LSIL to cervical cancer.

## Figures and Tables

**Figure 1 jcm-12-06188-f001:**
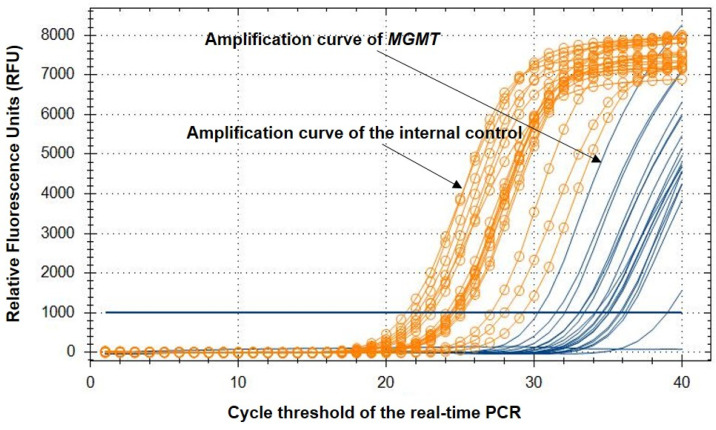
*MGMT* methylation analysis using the Epi-TOP MPP assay. *y*-axis: Relative fluorescence units (RFUs) of each fluorescence channel. *x*-axis: Cycle threshold of the real-time PCR. Orange curve (ROX fluorescence channel): amplification curve of the internal control (ACTB). Orange line: baseline threshold of the ROX channel. Green curve (HEX fluorescence channel): Amplification curve of the target gene (*MGMT*). Green line: baseline threshold of the HEX channel.

**Figure 2 jcm-12-06188-f002:**
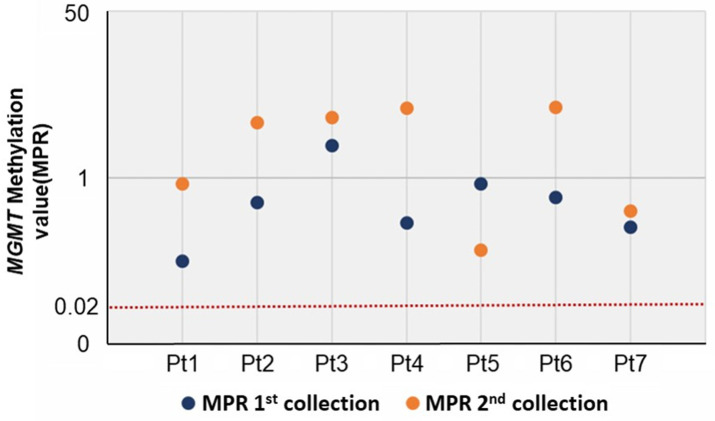
*MGMT* methylation values (MPR) of the 1st and 2nd collected samples from seven patients (defined unmethylated value: ≤0.02).

**Figure 3 jcm-12-06188-f003:**
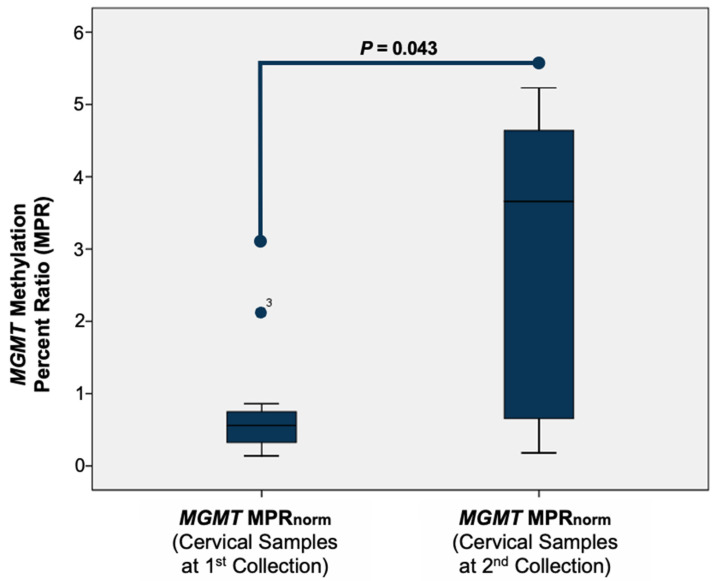
Normalization of the MPR of the samples at the 1st and 2nd collections using the Wilcoxon signed-rank test.

**Figure 4 jcm-12-06188-f004:**
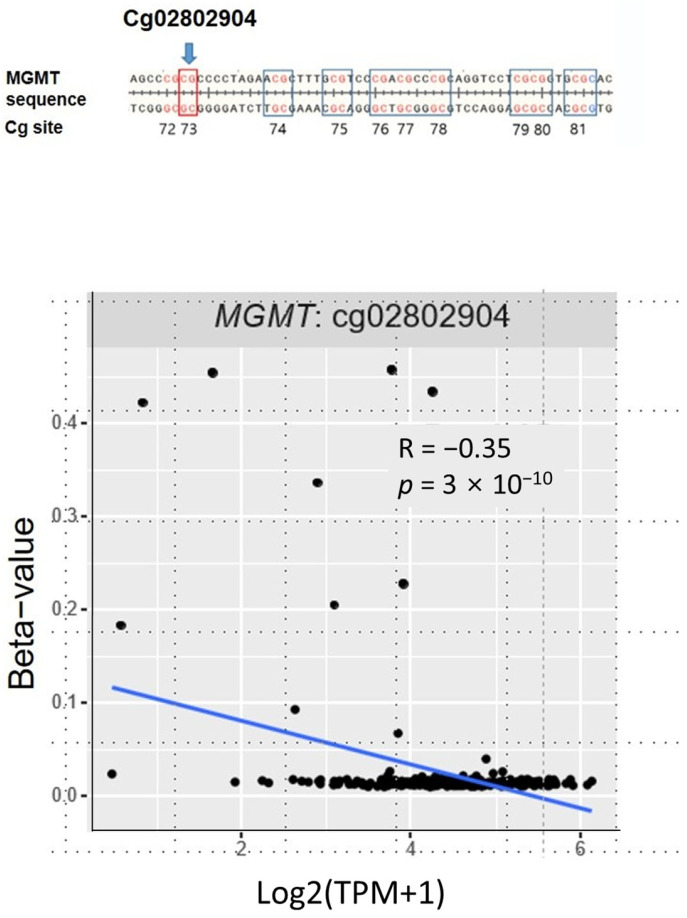
A negative correlation between the *MGMT* mRNA expression and MTMG methylation levels. Cg02802904, R = −0.35, *p* = 3 × 10^−10^. Statistical analyses were performed using Pearson’s correlation and Wilcoxon rank sum tests.

**Figure 5 jcm-12-06188-f005:**
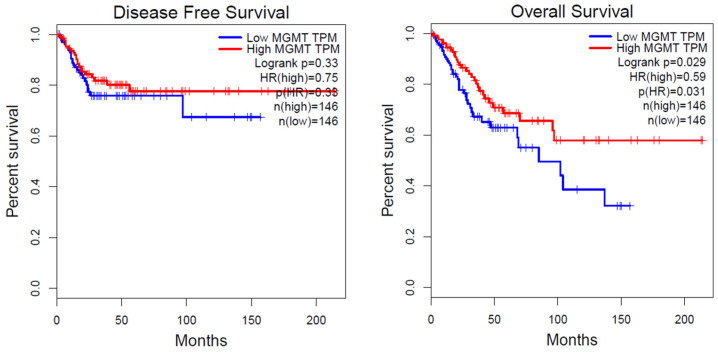
Relationships of the OS and DFS rates with *MGMT* mRNA expression. OS and DFS curves of *MGMT* mRNA expression using GEPIA based on profiles of primary cervical cancer from TCGA to allocate subjects into those with high/low gene expression, quartiles were used, *p* < 0.05.

**Table 1 jcm-12-06188-t001:** Clinical-pathologic characteristics of the patients.

Patient ID	Age	Sample Information	Cervical Cytology	HPV Types
Pt 1	32	1st collection	ASCUS	18
		2nd collection	NIL *	18
Pt 2	29	1st collection	ASCUS	33
		2nd collection	ASCUS	33
Pt 3	22	1st collection	ASCUS	39
		2nd collection	NIL *	39
Pt 4	58	1st collection	ASCUS	58
		2nd collection	ASCUS	58
Pt 5	49	1st collection	ASCUS	58
		2nd collection	ASCUS	58
Pt 6	37	1st collection	LSIL	52
		2nd collection	LSIL	52
Pt 7	41	1st collection	ASCUS	52
		2nd collection	NIL *	52

* NIL, negative for intraepithelial lesions or malignancy.

**Table 2 jcm-12-06188-t002:** *MGMT* methylation analysis data.

Patient ID	Sample Information	Ct (Control)	Ct (*MGMT*)	ΔCt	MPR
Pt 1	1st collection	27.29	36.82	9.53	0.14
	2nd collection	31.38	38.23	6.85	0.86
Pt 2	1st collection	27.93	35.41	7.48	0.56
	2nd collection	32.83	37.55	4.72	3.66
Pt 3	1st collection	31.06	36.58	5.52	2.13
	2nd collection	26.39	30.93	4.54	4.12
Pt 4	1st collection	24.21	32.39	8.18	0.34
	2nd collection	28.44	32.64	4.20	5.16
Pt 5	1st collection	23.21	30.06	6.85	0.86
	2nd collection	27.37	36.46	9.09	0.18
Pt 6	1st collection	27.47	34.75	7.28	0.64
	2nd collection	27.72	31.90	4.18	5.23
Pt 7	1st collection	21.74	30.05	8.31	0.31
	2nd collection	27.00	34.80	7.80	0.45

## Data Availability

No new data were created or analyzed in this study. Data sharing is not applicable to this article.

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
