# Peer review of "MGMT Methylation Is Associated with Human Papillomavirus Infection in Cervical Dysplasia: A Longitudinal Study"

_jcm, 2023, doi:10.3390/jcm12196188_

Round 1
Reviewer 1 Report
Commendable attempt to understand the molecular determinants of persistent HPV infection in the etiology of cervical cancer.
The limitations are well stated.
ABSTRACT
Abstract must be re-written; it fails to summarize the study well. Thus, one is left wondering how the authors plan to prove persistence of high-Risk HPV infection using studies related to ASCUS /LSIL. The information included from TCGA is out of place in its current form. This needs to either be removed or better integrated into the study.
Typographical error - delete ‘here from line ‘16’
‘TCGA data also revealed a negative association between MGMT methylation and mRNA expression in cervical cancer’ this sentence is better discussed in the discussion. I suggest it be removed from the abstract.
INTRODUCTION
This is well written
METHODS
This is well written
RESULTS
This is well written. However, the depiction in Fig 2 relating to patient 5 does not match with the conclusions drawn. The authors should provide reasons for the anomaly observed for patient 5 in relation to Fig 2.
DISCUSSION
Paragraph 4 ‘264 to 284’ should be re-worded. The argument to use MGMT methylation to determine a patients cervical disease status is one that is difficult to understand. A patient’s cervical disease is best determined by examining a cervical specimen. The patient cannot be said to have cancer when the cervix itself shows no evidence of it. It is not acceptable to assume that a patient has cancer without evidence of cancer in the cervix itself.
In relation to the same paragraph, the authors make a case about the low sensitivity of PAP in determining the presence of cervical dysplasia but fail to mention that it has a rather high specificity, the reason why it is used as a triage test after a positive HPV test. In this regard, unless the authors believe that MGMT methylation has equal or better specificity than than the PAP smear in determining cervical dysplasia, then their argument has to be reconsidered. HPV testing suffers from the same problem of low specificity with high sensitivity.
The last paragraph of the discussion appears to be a review of existing data with no relationship with the work done by the authors. It should either be deleted or presented as part of the literature. If the authors want to maintain it as part of the discussion then they should correlate it with their findings to ensure that it fits into the discussion.
There are typographical errors that need to be corrected.
Author Response
Comment 1 : ‘TCGA data also revealed a negative association between MGMT methylation and mRNA expression in cervical cancer’ this sentence is better discussed in the discussion. I suggest it be removed from the abstract.
Response 1: Thanks for the comment. I have removed the previous sentence based on the suggestion you provided. However, we added the sentences below to explain that our findings may suggest that the extent of MGMT methylation may hold significant potential as a prognostic marker for the progression from ASC-US/LSIL to cervical cancer;
“Moreover, the TCGA dataset revealed an association between downregulated MGMT mRNA expression and poor overall survival. This decreased MGMT mRNA expression was observed to have an inverse relationship with MGMT methylation levels. In this study, we found that the MGMT methylation level could potentially serve as a valuable prognostic indicator for the transition from ASC-US/LSIL to cervical cancer."
Comment 2: This is well written. However, the depiction in Fig 2 relating to patient 5 does not match with the conclusions drawn. The authors should provide reasons for the anomaly observed for patient 5 in relation to Fig 2.
Response 2: As the reviewer pointed out, our study indicated that the cervical samples from the 2nd collection generally exhibited higher levels of hypermethylation compared with those from the 1st collection. Figure 2 (below), as described in the figure legends, illustrated that all of the samples, including those from Patient 5, displayed hypermethylation, with a defined unmethylated value of ≤0.02. While it is indeed accurate that the cervical sample from the 1st collection was more hypermethylated than the one from the 2nd collection in Patient 5, our analysis of the normalized MPR level (MGMT MPRnorm) from all of the cervical samples revealed a correlation between the samples from the first and second collections (see Figure 3). We are unable to provide a definitive explanation for the reversal of results in Patient 5; however, it is important to note that our data appear reliable as the remaining six patients consistently exhibited higher levels of hypermethylation in their samples at the 2nd collection than the ones at the 1st collection.
Comment 3 : Paragraph 4 ‘264 to 284’ should be re-worded. The argument to use MGMT methylation to determine a patients cervical disease status is one that is difficult to understand. A patient’s cervical disease is best determined by examining a cervical specimen. The patient cannot be said to have cancer when the cervix itself shows no evidence of it. It is not acceptable to assume that a patient has cancer without evidence of cancer in the cervix itself.
In relation to the same paragraph, the authors make a case about the low sensitivity of PAP in determining the presence of cervical dysplasia but fail to mention that it has a rather high specificity, the reason why it is used as a triage test after a positive HPV test. In this regard, unless the authors believe that MGMT methylation has equal or better specificity than than the PAP smear in determining cervical dysplasia, then their argument has to be reconsidered. HPV testing suffers from the same problem of low specificity with high sensitivity.
Response 3: To begin with, we concur with the reviewer’s assertion that the most accurate method for determining a patient’s cervical disease status is through an examination of a cervical specimen. Consequently, in paragraph 4, we articulated that MGMT methylation might indicate the presence of HPV infection and could potentially serve as a biomarker for predicting persistent HPV infection (lines 259, 269–270). It should be noted that our intention was not to claim that MGMT methylation alone could definitively ascertain cervical disease status. Therefore, we have incorporated the below statements into paragraph 4 of our text. “Furthermore, addressing the specificity of HPV testing, recent studies have demonstrated that HPV-based screening offers results that are at least as reliable, if not superior, when it comes to predicting CIN 2+, CIN 3+, and invasive cervical carcinomas compared with traditional cytology-based methods [49, 50].” In addition, we rephrased the sentences in lines 261–286 to improve the readability.
Comment 4 : The last paragraph of the discussion appears to be a review of existing data with no relationship with the work done by the authors. It should either be deleted or presented as part of the literature. If the authors want to maintain it as part of the discussion then they should correlate it with their findings to ensure that it fits into the discussion.
Response 4: Thank you for the suggestion. We rephrased the paragraph as below.
" In this study, it was noted that, over time, ASC-US or LSIL samples with persistent high-risk HPV infections displayed a gradual increase in MGMT methylation levels. Additionally, we verified the significance of MGMT methylation in the progression of cervical cancer by analyzing public data from TCGA. Our findings suggest that the MGMT methylation level could potentially serve as a valuable prognostic indicator for the transition from ASC-US/LSIL to cervical cancer.”

Reviewer 2 Report
Choi et al investigated the methylation pattern of MGMT in cervical samples of 8 patients with ASCUS or LSIL. Their findings suggest that a) MGMT is hypermethylated in ASCUS and LSIL samples in comparison with normal samples b) MGMT hypermethylation is associated with persistent HPV infection. They also analyzed TCGA data to show that MGMT mRNA expression is inversely correlated to methylation status and that low expression of MGMT is associated with poor overall survival of cervical cancer patients.
The correlation of MGMT methylation LSIL, HSIL and cervival cancer has been the topic of several studies. The most interesting finding of the current study is the link between MGMT methylation and persistent HPV infection. However, the presented data do not fully support the conclusions of the paper.
Specific points:
1. The cohort contains too few patients.
2. What are the control samples? Additionally, if ASCUS and LSIL samples are MGMT hypermethylated, how do the authors explain the higher MGMT methylation of NIL samples from patients 1,3 and 7? In all these patients, MGMT MPR is higher in NIL samples than ASCUS samples. This contradicts the conclusion in lines 273-274.
3. The correlation between MGMT methylation and persistent HPV infection is too preliminary because of the size of cohort and the absence of a suitable control group.
4. Section 3.3 refers to MGMT as a prognostic and not a predictive biomarker.
The manuscript needs editing for clarity.
Author Response
Comment 1 : The cohort contains too few patients.
Response 1: We appreciate your comment. Despite our study’s small sample size, consisting of only seven patients, we conducted DNA analysis on cervical lesion samples from each patient at two distinct time points. As detailed in the manuscript, obtaining longitudinal DNA data is a complex task. Therefore, even with the limited number of patients in our study, our findings hold significant value.
Comment 2: What are the control samples?
Response 2: As written in the materials and methods, the manufacturer used “normal uterine tissues” to define MGMT’s unmethylated state.
Comment 3: Additionally, if ASCUS and LSIL samples are MGMT hypermethylated, how do the authors explain the higher MGMT methylation of NIL samples from patients 1,3 and 7? In all these patients, MGMT MPR is higher in NIL samples than ASCUS samples. This contradicts the conclusion in lines 273-274.
Response 3: In our concluding remarks (lines 273–274), we emphasized that MGMT methylation might serve as a biomarker for predicting persistent HPV infection, rather than relying solely on PAP smear results. As discussed earlier, the PAP smear test is associated with a significant false negative rate. Therefore, the authors underscored the significance of utilizing a biomarker for accurate prediction of persistent HPV infection.
Comment 4 : The correlation between MGMT methylation and persistent HPV infection is too preliminary because of the size of cohort and the absence of a suitable control group.
Response 4: Thank you for the comment. First, we acknowledge that the sample size is small, and we have highlighted this in our discussion. As indicated in response 1, even though our sample size was limited to seven patients, we conducted DNA analysis on cervical lesion samples from each patient at two distinct time points. Furthermore, given the inherent challenges of acquiring longitudinal DNA data, the significance of our findings should not be underestimated, despite the modest patient enrollment. Second, we recognize the absence of an appropriate control cohort in our study. However, we took measures to address this limitation by utilizing control samples outlined in the materials and methods section. Specifically, the manufacturer employed “normal uterine tissues” as a reference for defining the unmethylated state of MGMT.
Comment 5 : Section 3.3 refers to MGMT as a prognostic and not a predictive biomarker.
Response 5: In accordance with your suggestion, we revised section 3.3 to “MGMT as a predictive biomarker.”

Round 2
Reviewer 2 Report
The authors addressed some of the reviewers' concerns and the manuscript is improved. However, there are issues that need to be addressed before publication.
a)The statement that MGMT methylation may be a prognostic indicator of the transition from ASCUS-LSIL to cervical cancer is not supported by their data and should be removed. The same applies to the statement that MGMT methylation status may help identifying the patients' status (lines 35-36 of the abstract and lines 270-276).
b) The manuscript has two parts; the first one that correlates MGMT methylation status with HPV infection (the authors should mention the method that was used to detect HPV) and the second presenting the prognostic role of MGMT methylation in cervival cancer patients. These are two separate analysis and it should not be assumed that MGMT methylation status predicts the transition of intraepithelial neoplasia to cancer.
c) The current version lacks figures. Are the figures the same with the original version?
c)Regarding the title and the repeated mention of intraepithelial neoplasia in the manuscript: Most of the patients in this cohort have ASCUS instead of LSIL therefore the constant mention of intraepithelial neoplasia is not very appropriate. The current findings should be interpreted in line of this PAP smear findings.
d) The current version lacks figures. Are the figures the same with the original version?
Author Response
September 19th, 2023
Editors, Journal of Clinical Medicine
Dear Mrs. Camelia-Adina Morovan,
On September 14th, a decision letter for the manuscript, jcm-2570827, entitled “MGMT methylation is associated with human papillomavirus infection in cervical intraepithelial neoplasia: a longitudinal study” was received. The manuscript has been reviewed and revised in accordance with the reviewers’ comments. We obtained a supplementary English editing service to improve the overall clarity. The corrected sections in the revised manuscript are marked in yellow.
Reviewer #3>
Comment 1: The statement that MGMT methylation may be a prognostic indicator of the transition from ASCUS-LSIL to cervical cancer is not supported by their data and should be removed. The same applies to the statement that MGMT methylation status may help identifying the patients' status (lines 35-36 of the abstract and lines 270-276).
Response 1: Your commands are appreciated. It is important to note that our assessment of MGMT methylation status in cervical cancer relied on publicly available data (TCGA), rather than our own data. Nevertheless, it is worth mentioning that subsequent to the release of TCGA data, numerous studies have utilized this public dataset for validation purposes. In a study conducted by Farshidfar F et al., it was observed that ARID1A exhibited DNA hypermethylation and decreased expression levels in the IDH mutant subtype associated with cholangiocarcinoma (CCA). These findings were gleaned from the analysis of TCGA data [1]. Another investigation led by Nagy A et al. involved an extensive search through the TCGA (RNA-seq) and GEO (microarray) databases in pursuit of miRNA datasets encompassing both expression and clinical information. Remarkably, within the RNA-seq and gene chip datasets, 55 and 84 miRNAs, respectively, were found to exhibit significant correlations with overall survival. Moreover, various studies have independently confirmed the methylation patterns of specific genes utilizing TCGA data [2].
We want to clarify that our assessment of MGMT methylation in cervical cancer relied on data from the TCGA dataset, rather than our own. Consequently, we underscored the importance of considering MGMT methylation level as a potential prognostic biomarker, rather than a definitive one. Therefore in the concluding section of the abstract we wrote as below; "Moreover, the TCGA dataset revealed an association between downregulated MGMT mRNA expression and poor overall survival. This decreased MGMT mRNA expression was observed to have an inverse relationship with MGMT methylation levels. In this study, we found that the MGMT methylation level could potentially serve as a valuable prognostic indicator for the transition from ASC-US/LSIL to cervical cancer.".
In addition, in lines 270-276, we articulated that MGMT methylation might indicate the presence of HPV infection and could potentially serve as a biomarker for predicting persistent HPV infection. It should be noted that our intention was not to claim that MGMT methylation alone could definitively ascertain cervical disease status. Therefore, we changed lines 270-276 to the following: As mentioned above, HPV status may change over time and PAP smear may produce false negative results, so in such cases MGMT methylation analysis may help identify the patients’ cervical disease status (NIL, precancer or cancer). Second, MGMT methylation might serve as a biomarker for predicting persistent HPV infection. HPV DNA testing and PAP cytology provide information about the disease status at the time of analysis, but do not offer predictive insights into which patients with high-risk HPV infections will remain persistently infected or progress to cervical cancer. Currently, there are no markers available for predicting HPV infection persistence or the development of cervical cancer. Our study has shown an association between MGMT methylation and persistent HPV infection, suggesting that MGMT methylation may have a role as a biomarker for predicting persistence in such cases.
Comment 2: The manuscript has two parts; the first one that correlates MGMT methylation status with HPV infection (the authors should mention the method that was used to detect HPV) and the second presenting the prognostic role of MGMT methylation in cervical cancer patients. These are two separate analysis, and it should not be assumed that MGMT methylation status predicts the transition of intraepithelial neoplasia to cancer.
Response 2: We concur with the reviewer's assessment that there are two distinct analyses to consider; one involving MGMT methylation and HPV infection, and the other involving MGMT methylation in the prediction of cervical precancer or cancer.
In paragraph 4, we articulated that MGMT methylation might indicate the presence of HPV infection and could potentially serve as a biomarker for predicting persistent HPV infection. It should be noted that our intention was not to claim that MGMT methylation alone could definitively ascertain cervical disease status. Therefore, we changed the sentences as following: As mentioned above, HPV status may change over time and PAP smear may produce false negative results, so in such cases MGMT methylation analysis may help identify the patients’ cervical disease status (NIL, precancer or cancer). Second, MGMT methylation might serve as a biomarker for predicting persistent HPV infection. HPV DNA testing and PAP cytology provide information about the disease status at the time of analysis, but do not offer predictive insights into which patients with high-risk HPV infections will remain persistently infected or progress to cervical cancer. Currently, there are no markers available for predicting HPV infection persistence or the development of cervical cancer. Our study has shown an association between MGMT methylation and persistent HPV infection, suggesting that MGMT methylation may have a role as a biomarker for predicting persistence in such cases.
Comment 3: Regarding the title and the repeated mention of intraepithelial neoplasia in the manuscript: Most of the patients in this cohort have ASCUS instead of LSIL therefore the constant mention of intraepithelial neoplasia is not very appropriate. The current findings should be interpreted in line of this PAP smear findings.
Response 3: Thank you for your insightful comment. In response to your commentary, the terminology 'intraepithelial neoplasia' will be amended to "cervical dysplasia”. Furthermore, numerous studies have grouped ASCUS and LSIL in the context of cervical dysplasia because they are with low-grade cytology test results. One study even demonstrated that LSIL and HPV positive ASCUS have comparable clinical significance in predicting CIN2/3 [3,4].
Comment 4: The current version lacks figures. Are the figures the same with the original version?
Response 4: We submitted five figures in accordance with the conditions generally required by journals. However, it is worth noting that a recent publication in JCM featured nine figures, while numerous others contained fewer than five.
Reference
- Farshidfar F, Zheng S, Gingras MC, Newton Y, Shih J, Robertson AG, Hinoue T, Hoadley KA, Gibb EA, Roszik J, Covington KR, Wu CC, Shinbrot E, Stransky N, Hegde A, Yang JD, Reznik E, Sadeghi S, Pedamallu CS, Ojesina AI, Hess JM, Auman JT, Rhie SK, Bowlby R, Borad MJ; Cancer Genome Atlas Network; Zhu AX, Stuart JM, Sander C, Akbani R, Cherniack AD, Deshpande V, Mounajjed T, Foo WC, Torbenson MS, Kleiner DE, Laird PW, Wheeler DA, McRee AJ, Bathe OF, Andersen JB, Bardeesy N, Roberts LR, Kwong LN. Integrative Genomic Analysis of Cholangiocarcinoma Identifies Distinct IDH-Mutant Molecular Profiles. Cell Rep. 2017 Mar 14;18(11):2780-2794.
- Nagy, Á., Lánczky, A., Menyhárt, O. et al.Validation of miRNA prognostic power in hepatocellular carcinoma using expression data of independent datasets. Sci Rep 8, 9227 (2018)
- Walker JL, Wang SS, Schiffman M, Solomon D; ASCUS LSIL Triage Study Group. Predicting absolute risk of CIN3 during post-colposcopic follow-up: results from the ASCUS-LSIL Triage Study (ALTS). Am J Obstet Gynecol. 2006 Aug;195(2):341-8
- Cox JT, Schiffman M, Solomon D; ASCUS-LSIL Triage Study (ALTS) Group. Prospective follow-up suggests similar risk of subsequent cervical intraepithelial neoplasia grade 2 or 3 among women with cervical intraepithelial neoplasia grade 1 or negative colposcopy and directed biopsy. Am J Obstet Gynecol. 2003 Jun;188(6):1406-12